# Teaching How to Research: A Case Study on Chemical and Industrial Engineering Degrees

Sergio Nogales-Delgado [1],*, José María Encinar Martín [2] and Silvia Román Suero [1]

1   Department of Applied Physics, University of Extremadura, Avda. de Elvas s/n, 06006 Badajoz, Spain
2   Department of Chemical Engineering and Physical-Chemistry, University of Extremadura, Avda. de Elvas s/n, 06006 Badajoz, Spain
*   Correspondence: senogalesd@unex.es

**Abstract:** Final Degree Projects (FDPs) in scientific and technical studies are often the first significant approach to real research and can be a challenging step for students. Their own experience at this stage can influence the students' professional calling to a research career instead of other technical occupations. In any case, it should be an interesting experience for students, regardless of their future career. Thus, suitable guiding by academic tutors is vital to favor the engagement for scientific research as a feasible professional career in youngsters. The aim of this work was to point out our own experience when it comes to teaching how to research, inspired by research-based learning. In other words, this is an example about how to introduce final degree students to research, to prepare them for a possible future career in the research world. Specifically, the expectation for each FDP was to materialize their work in the publication of a research paper or a conference communication, in the context of research carried out in the frame of a transference project with a firm purpose: to emphasize how their work will be applied as a solution to a real problem. To achieve this goal, a general didactic guide (which should be customized for each student) is presented, adapted to a chemical engineering laboratory, mainly devoted to biodiesel and biolubricant production.

**Keywords:** final degree project; English for science; biorefinery; conference communication; scientific article; problem-based learning; research-based learning; university education; STEM; engagement





## 1. Introduction

### 1.1. Research as a Key Strategy of States: The Role of Education

The relevance of research to improve technology development and the subsequent economic growth in many aspects is well known. In the case of research devoted to green approaches, profitability is not only economic but also social and becomes more and more supported by important policies [1]. Particularly during the last decade and due to the energy crisis, climate change and severe pollution problems, countries all over the world have increased their efforts to research renewable energy from more efficient sources, green sustainable products and processes to lower carbon and hydric footprint, improving energy storage devices.

In this line, developed countries usually devote a considerable percentage of their Gross Domestic Product (GDP) in research and development budgets, proving that a continuous increase in research and technology investment is usually related to global economic growth. Compared to other countries, such as Israel, Germany, USA or China (whose R&D investment represents around 5, 3, 2.8 and 2.2% GDP in 2018), Spain still has a long way to go to achieve similar figures in research investment, as the current GDP percentage devoted to research is around 1.25%, not showing a considerable increasing trend in the last years [2], although a long-term commitment to increase R&D investment up to 3% has been proposed by the Spanish Government in the frame of the European Green Deal.

From this data, many remarks can be formulated concerning the Spanish situation:

- There is an urgent need to promote research practices and improve the investment in research, in order to broaden the amount of research lines and their quality.
- In the long term, there seems to be an opportunity to create a labor and vocational niche, which should be filled with professionals and researchers from a wide range of disciplines (not only scientific or technical ones).

As a result, all educational institutions play a vital role in order to promote research culture at all educational levels. In the case of universities, which typically represent the last step for students before their first professional experience, this role is significantly more important, as it is the right place where education and practical research usually go together, requiring transversal skills applied to working environments [3,4]. Further, higher-education institutions have a strong influence on students' sense of belonging and commitment to their professional success [5,6]. Universities can serve as an inspiration for defining the vocation of future graduates and, in that sense, the promotion of research and scientific culture should be intensified in order to obtain positive outcomes (that is, professionals devoted to R&D). Although the near future seems "promising" when it comes to research (which is an essential task for universities and technological centers), the current situation offers positive and negative aspects, with some interesting challenges, such as the decrease in students enrolled at universities, especially in engineering degrees [7].

In order to understand the importance of teaching research skills, a review of the evolution of students enrolled at Spanish Universities (at a national and regional level) was conducted (see Figure 1). In general, enhanced interest was found for technical disciplines, such as engineering and architecture, at all educational levels (for instance, there were around 250,000 engineering students at university in the last decade, whereas the amount of scientific students never surpassed 9000 in this period). These differences between the number of engineering and science students were especially noticeable for master's degrees (with an engineering/science student ratio of around 4.5 at national and regional level in course 2021–2022). As it can be observed, there is an increasing number of student enrollments in master's studies devoted to engineering and architecture, which can be explained by the fact that these disciplines are eminently practical and master's are employment oriented, generally focused on practical contents. Surprisingly enough, more engineering students opted to enroll in PhD degrees, at a national level, at the expense of scientific disciplines, such as chemistry, physics, mathematics, etc., traditionally with a wide range in PhD programs. Nevertheless, in our region, the opposite took place and scientific PhDs tripled against engineering PhDs in the course of 2021–2022.

This fact shows a clear trend towards practical results, possibly due to new teaching practices, such as PBL or RBL (problem- or research-based learning),as in the research and development projects developed by universities, companies are playing a more and more important role by means of transference-oriented collaboration agreements. On the other hand, although the number of degree university students decreased recently (showing a slight stabilization and a posterior increase from course 2019–2020 at a national level), the interest in postgraduate studies has been steadily increasing, as in the case of PhD degrees.

Focusing on the case of the Extremadura region, which has one of the lowest percapita incomes in Spain and Europe, the need for research is vital in order to promote sustainable economic growth, especially related to its agricultural tradition and natural resources. In that sense, research related to engineering and architectural disciplines is not remarkable, compared to a national level, and the number of PhD students remained stable and low during the last seven courses. On the other hand, the increase in master's students is not equally observed at a regional level, as there was not a steady increase during the same academic period. Finally, the decrease in university students in engineering and architecture degrees could contribute to a negative trend in the future for PhD degrees.

Within this context, the role of final degree projects (FDPs), where the student is considering future professional and educational alternatives, could be vital, implying an important starting point to introduce, promote and reinforce scientific culture in practice,

which could contribute to an increasing interest for graduate students in research. In other words, FDPs could contribute to a steady engagement of students for research.

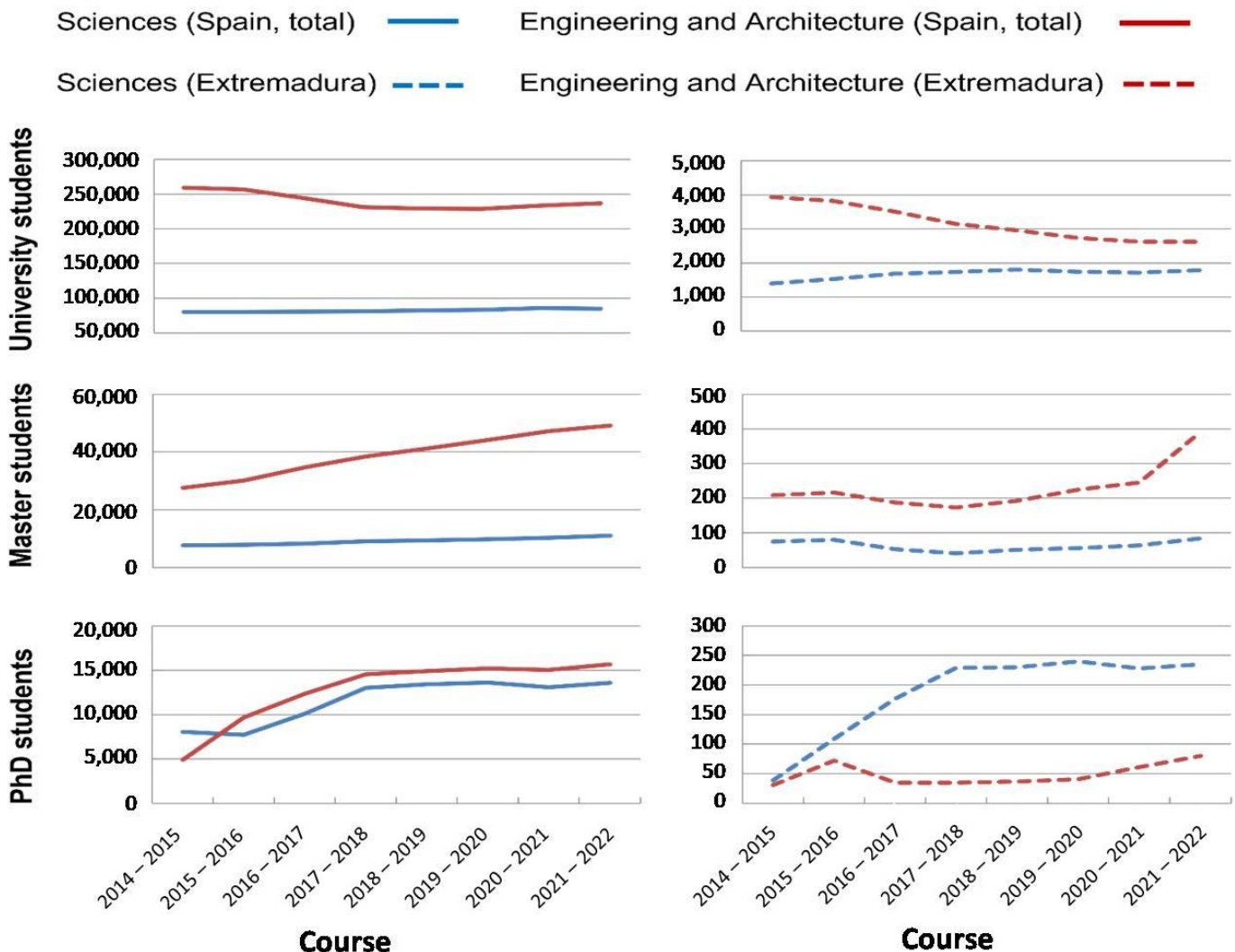

**Figure 1.** Evolution of students enrolled in Spain and Extremadura at different educational levels during the last seven academic years [8].

### 1.2. Final Degree Projects: Joint Knowledge Gathering and Maturity Push

Final degree projects (FDP) are usually the last step in a degree and consist of a written project and an oral presentation. Making these projects involves an effort to synthesize and associate diverse specific information acquired by the student during the degree (conceptual, practical and academic skills) but, in addition, will expose the student to a challenge in which he/she will have to manage times, gain self-confidence, tackle adverse situations or make a planning of resources, apart from learning many other transversal skills [9]. During the development of FDPs, the students face both material and human responsibilities, learn (or improve) how to work in groups and share spaces (this is particularly important for experimental FDPs, where several students work in a laboratory and use the same equipment or chemical products) and deal with stress [10].

Above all this, FDPs have a very significant role on the student perception of his/her ability to solve a real problem and the way it is developed can be decisive on the motivation or engagement of the student. In other words, problem-based or research-based learning could be an interesting tool to encourage students at all educational levels.

The term "motivation" has been addressed quite often in relation to FDPs. Danowitz et al.investigated how an FDP can help students enthuse over their studies and help them understand the meaning of their future role as professionals; this author has even suggested that FDPs could be defined and started during the first courses (not at the end) and be developed gradually, to enhance student enthusiasm in his/her learning process [11].

### 1.3. Research-Based Learning as a Motivation Booster

Thus, research-based learning (RBL) is becoming more and more important nowadays, not only in STEM areas, but in various and different subjects, such as tourism education [12], or other kinds of audiences, such as the general public, in scientific exhibitions [13]. Indeed, every discipline requires the engagement of students and RBL seems to be one of the most effective ways to face fast-changing realities in learning environments. In this way, most RBL courses or projects (regardless of the subject) usually offer these points in common, among others [14]:

- RBL fosters autonomy and independence in students.
- There is a noticeable involvement in the research process by students.
- It usually takes time to adapt to this process (for teachers to elaborate RBL programs; for students, to get used to this methodology).
- These practices ensure rigor and critical thinking.
- Research and teaching usually imply a positive and synergistic effect on student's academic performance.

As a consequence, a student's involvement implies better acquired skills, preparing them for their future career. Consequently, many institutions tend to foster this kind of approach (combining research and teaching) through different strategies and, in this case, universities play a vital role, as explained earlier [15].

However, how teaching and research should be jointly accomplished might vary according to different factors, such as the subject to be adapted, mentor or teacher's points of view, the kind of approach of teaching, etc. [15]. In that sense, some studies about the possible implementation of research-based learning (also problem-based learning), combining laboratory work (or research projects) with academic teaching, have been carried out recently in STEM education. Thus, biochemistry and medicine seem to be interesting fields to implement these kinds of methodologies [16]. For instance, an undergraduate biomedical laboratory course has been implemented continuing in this vein, improving students' learning and academic skills due to their confidence and involvement in research [17]. Other courses are based on a single problem in order to introduce different specific concepts, such as learning nucleic acid biochemistry from AIDS research or learning about wild-type and mutant clones of malate dehydrogenase. Again, students developed a sense of confidence and involvement, improving their academic results and motivation [18,19].

Nonetheless, and although some of these programs have some points in common (as explained above), their specificity requires further studies about different and diverse fields, such as chemical and technical engineering (which is our field of expertise), in order to fill these gaps in knowledge. Our previous works tried to deal with different educational aspects related to chemical and technical engineering, including safety courses or didactic guides for science exhibitions [20,21]. In a sense, these works were the background or foundations for this educational guide, as we tried to combine research and education as much as possible, with the inestimable collaboration of our students. Thus, the novelty of this work is the application of research-based learning to final degree projects in chemical and technical engineering, trying to adapt the foundations of this method to these specific fields (which are not widely considered in the literature).

Considering the above, the aim of this work was to propose a methodology (mainly based on our own experience, but also considering previous works) to make research in engineering subjects more attractive for final degree students, in order to promote this discipline as a feasible career, rather than the usual technical professional careers. Further,

preliminary results about the satisfaction and career opportunities of the students were included. Thus, we will try to answer the following questions:

- How can this method contribute to a higher commitment to research?
- Did this work improve the results dissemination of our department?
- What is the difference between this method and the previous ones concerning research-based learning?

## 2. Context and Methodology

Once the situation has been analyzed at a glance, the specific scope is detailed next, based on the main characteristics of our educational experience:

- The laboratory where the educational experience takes place belongs to the Department of Chemical Engineering and Physical-Chemistry and the Department of Applied Physics of the University of Extremadura and it is devoted to subjects and research lines related to Chemical and Technical Engineering.
- The staff includes professors, university teachers and scientific staff, who can act as mentors in final degree projects (FDP). Thus, there are two professors, three university teachers and five researchers (although professors and teachers also contribute to research), who usually participate in pairs for each FDP. All of them have a considerable experience in research (especially professors), with at least a two-year stay in this laboratory. In that sense, the use of a guide would be useful to homogenize the pedagogical content included in FDPs. All the FDPs taking place in this laboratory took part in a research project devoted to biofuels, including subjects, such as biomass, biodiesel and biolubricant production. Thus, the main purpose of this project was to foster renewable energies in our region, but there is a specific part whose aim was to promote research practices at the university level, making students a part of this project. That is the origin of this work, as we tried to assess the impact of research-based learning applied to renewable energies on future chemical and technical engineers. In this way, the FDP of each student was a part of a more complex project, but it was (in essence) their own work, gaining as much autonomy as they could.
- The main subjects explained in this branch of the department are related to Chemical and Technical Engineering, especially devoted to reaction mechanisms and kinetics during conversion reactions, in the context of energy and fuels.
- Apart from the abovementioned research project, the main research lines are related to energy from biomass, biofuels (mainly biodiesel) and bioproducts (such as glycerol or biolubricants), along with catalyst characterization, among others.

In this context, one of the main tasks carried out in our laboratory is the development of final degree projects (FDP) for Chemical and Technical Engineering students. In this study, 16 male and 16 female students were included, whose ages ranged from 22 to 28 years, depending on their ease in completing their corresponding degrees. This group of students was compared with previous students who did not enjoy this methodology (from 2006–2007 to 2013–2014 academic year, 21 male and 14 female students, whose age range was 23–26 years). In Table 1, the evolution of FDP students during the last seven years is included:

**Table 1.** Evolution of FDP students in our laboratory.

| Academic Year | Total FDP Students | Subject | |
|---|---|---|---|
| | | Biorefinery | Combustion and Pyrolysis |
| 2014–2015 | 2 | 2 | 0 |
| 2015–2016 | 3 | 2 | 1 |
| 2016–2017 | 3 | 3 | 0 |
| 2017–2018 | 4 | 3 | 1 |
| 2018–2019 | 5 | 3 | 2 |
| 2019–2020 | 5 | 4 | 1 |
| 2020–2021 | 3 | 2 | 1 |
| 2021–2022 | 7 | 5 | 2 |

Considering courses from 2014–2015 to 2021–2022, the increase in students in 2017–2018 was consistent with data related to the trend found at the regional level (see Figure 1).

We believe that in view of the amount and increasing trend of students making FDP in our research area, making a guide gathering specific and transversal skills associated to this education level could be suitable. This guide would be useful to optimize the experience of both students and teachers during the FDP, especially concerning the following:

- Educational quality.
- Efficient time management.
- Enhancement of student transversal skills: autonomy and self-motivation
- Academic grade.
- Scientific production.

Apart from designing and implementing the FDP research guide, this work also covers the tracking of students and detection of lacks during their training, as well as carrying out a follow-up of graduate students in order to assess their professional choices after this educational experience.

The didactic guide has been clearly inspired by the mentoring process and research-based learning, which is highly adequate in this context. Obviously, this guide was mainly based on our own experience, as it usually happens to other mentors in disciplines such as teacher education [14].Through this approach, the role of the mentor and mentoree changes during the several stages of the learning process, especially concerning the degree of supervision by mentors and the degree of freedom of choice by mentorees.

In this way, at early stages the student has less freedom of choice, as they usually apply for FDPs that are previously established by the department, according to their needs. Further, the supervision of the student is stricter, as they need a training period to acquire laboratory and research skills. In this period, we pay attention to different aspects to provide additional transversal skills, such as the implementation of a complete preventive culture, by applying safe practice in the laboratory.

At this moment, we also give information about the research context and focus on the challenging situation the world is facing regarding the shortage of energy supplies, lack of food, drought and climate migration movements and our responsibilities as consumers in the world. At this point, we like to give a certain degree of freedom to the student, to really understand how resources, politics and human rights can be related and to appreciate the importance of working in biofuels; we encourage him/her to devote one project chapter to these considerations and to navigate on world indexes related to economics, energy, geopolitical strategies and so on [22].

Once the student gains experience, the supervision becomes lighter and the student becomes more independent; this enhanced "freedom" relates to different aspects: they have free choice to search literature and select data to be included in their final degree report, which can be consulted to the mentor in their regular meetings. Further, they can even change some aspects of the experimental design if necessary, with prior approval of the tutors. In other words, the mentoring process achieves its fullness and the mentor guides the mentoree according to the decisions made by the latter. The gradual autonomy that the supervisor gives to the student is described next in more detail, through the different steps. In the same line, several researchers have established different increasing levels where the student usually gains independence and freedom when it comes to decision-making in research-based education [23,24].

### 2.1. First Meeting

This is one of the most important stages during the FDP process, being the first contact between the tutors and the student. In this meeting, the main goals of the FDP are exposed and the tutors know the main particularities of the student (time availability, knowledge about basic aspects such as scientific English level, if they have any pending subjects in the degree, etc.). Based on the information gathered at this first meeting for our students (for eight consecutive courses, included in Table 1), their skills and personal circumstances are evaluated. They were

asked about different aspects that we consider important to carry out an FDP properly, including the following items: references (for instance, how to cite an author), English level, writing and laboratory skills and others (where the students can add any particularity that can affect the correct development of their FDP). From our results, the main aspects that we should focus on (to carry out the final degree project properly) can be drawn out in Figure 2, where the percentage of students having problems with each subject is included.

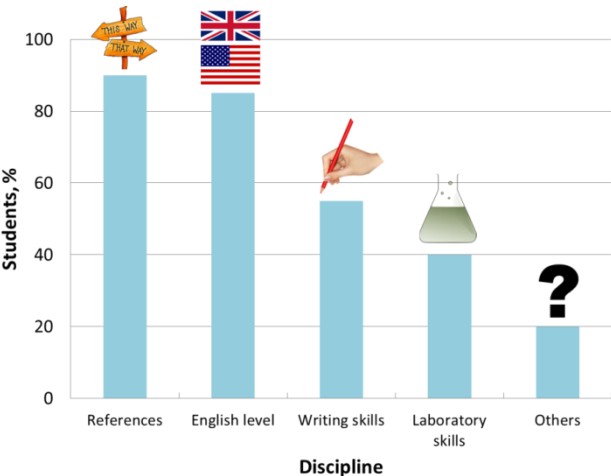

**Figure 2.** Main concerns related to the development of FDPs in our laboratory.

As inferred from this figure, all the students covered in this study had problems about how to reference their future work. Thus, the use of reference programs (such as Zotero or Mendeley) is vital. Other aspects such as their English level and writing skills were also considerable, even though they had to write technical reports in previous courses. Regarding laboratory skills, most of them had carried out laboratory practices, although the specific nature of this work is unknown to them. Finally, other concerns, related to their previous knowledge about the subject of the FDP or its feasibility, affected our students to a lesser extent. Depending on the information provided by the students, paying attention to their main concerns, the tutors provide a customized literature list. Table 2 shows typical bibliographical sources we give the students to facilitate their work.

**Table 2.** Recommended literature for the main aspects of training.

| Concept | Recommended Literature (References) |
|---|---|
| Scientific English | Books [25,26] |
| Literature research | University website [27] |
| Nomenclature | IUPAC [28] and books [29] |
| Organic Chemistry | Books [30,31] |
| Pyrolysis/combustion * | Reviews, our articles [32–34] |
| Biodiesel | Reviews [35–38], own work [39–41] |
| Glycerol | Reviews, own work [42] |
| Biolubricants | Reviews [43–46], own work [47–50] |
| Biorefineries | Reviews [51–53] |
| Antioxidants | Reviews [54–56], own work [49,57,58] |
| Oxidative stability | Reviews [58,59], own work [57,58,60] |
| Viscosity | Own work [40,48,61] |
| Our laboratory work | Own work [62–65] |
| Risk prevention course | Own procedures [20] |
| Exhibition guide ** | Own procedures [21] |
| CLIL class ** | Own procedures (not officially published yet) |
| Evaluation criterion | University website [66] |

* Especially for students from Industrial Engineering School. ** If the FDP students become a doctoral candidate who can teach in some courses.

The bibliographic sources are adapted to the previous knowledge of our students, who mainly come from Chemical or Technical Engineering or other degrees such as Chemistry or Environmental Management. Master's such as Industrial Engineering or Renewable Energy are also sources of FMP (Final Master's Project) students in our area. Nevertheless, all projects have some common topics. For instance, safety at lab, literature search or scientific English are common disciplines that should be mastered by students. Thus, Figure 3 shows a summary of the main contents and the nature of the student.

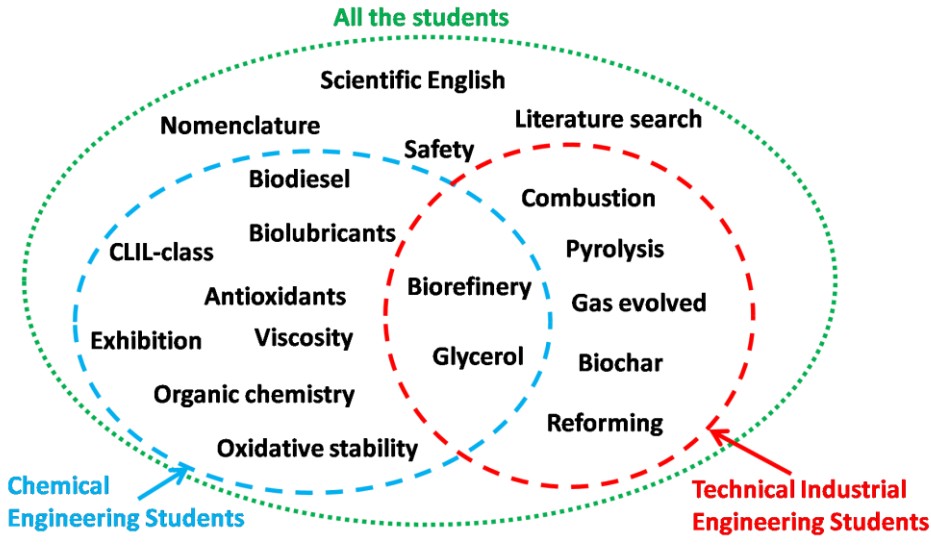

**Figure 3.** Circle packing about the main concepts covered in the FDP depending on the university degree of the student.

As can be seen, depending on the skills of the student, we will focus on some aspects at the expense of others. For instance, for students from the Industrial Engineering School, whose main research (related to our laboratory) is combustion and pyrolysis of biomass, we will provide them a literature focused on these fields, rather than biodiesel or biolubricant synthesis (which is usually suitable for chemical engineering students). On the other hand, other subjects can be shared by Chemical and Technical Engineering students, such as those related to biorefineries or glycerol. This fact points out the possibility of adaptation of this method to other subjects in STEM disciplines (as it is a matter of including a new circle in this figure, with their specific and common contents).

This "library" is not a static database provided to the students in order to study the main aspects related to the FDP. It is continuously changing and evolving, as students have a vital role in the review process, which is one of the most important stages in final degree projects, in our opinion. It is vital for the renewal of data files and reference lists. Additionally, these kinds of tasks (among others) usually promote engagement in students, as they feel part of the research project and they know that their contribution will remain. Accordingly, as explained elsewhere, engaging students in research activities and curricula is vital and both students and universities obtain substantial benefits, as we will see in following sections [15].

In this way, we encourage students to start reading the recommended literature shown in Table 2 (adapted to their field of study), obtaining basic information (collected in the students' most preferred format) and new interesting references, starting the process again. The new references obtained should be stored in a database (such as Mendeley, Zotero or Refworks) for further citations. Thus, this is one of the main contributions of the student to our laboratory and for the upcoming students, who will appreciate an updated and complete set of documents.

### 2.2. The Adoption of a Schedule

After the first meeting, where the main needs of the student are also considered (for instance, the possibility of defending the FDP as soon as possible, future professional expectations, etc.), a schedule is designed and proposed to the student, based on previous experiences where tutors or mentors had the opportunity to assess the average duration of each proposed task. In that sense, the schedule is usually designed to foresee future eventualities during the development of the FDP (for instance, time off work of tutors, or confinements due to COVID-19 outbreak, among other countless reasons), adding "extra time" for each task to ensure that all of them are accomplished on time. Figure 4 shows an example of a schedule, with the main tasks disaggregated in other specific tasks.

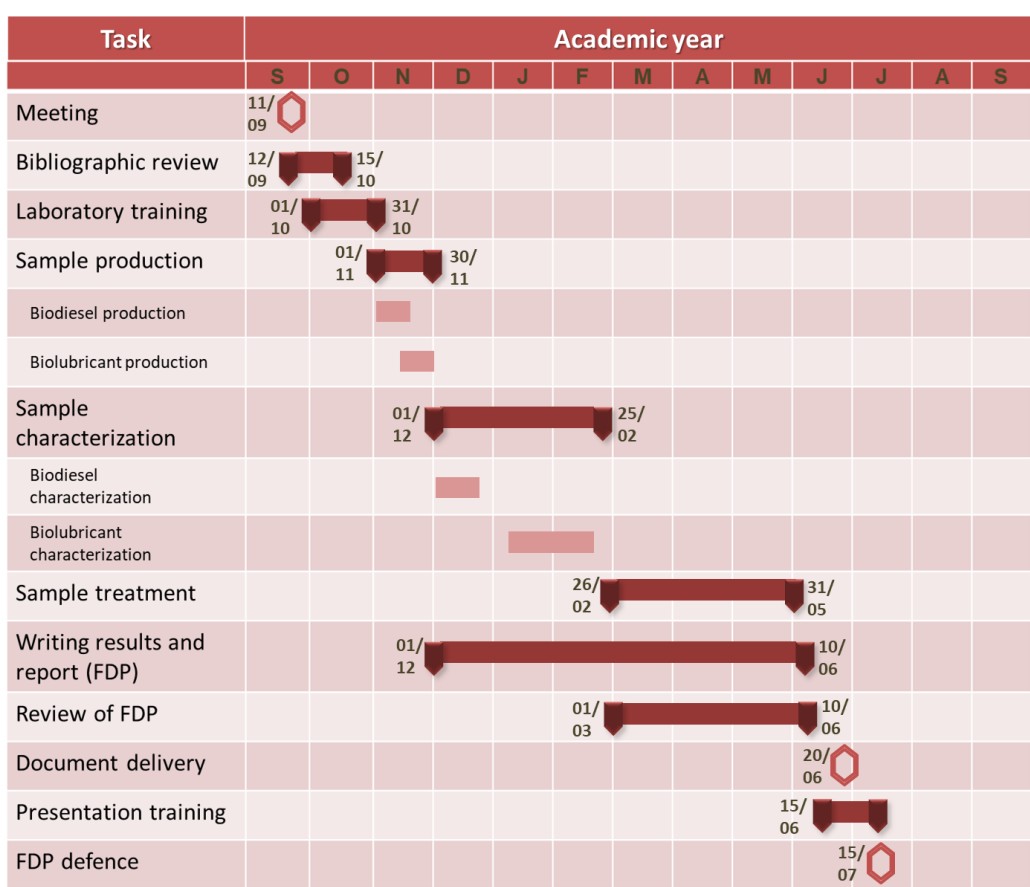

**Figure 4.** Typical example of a general schedule.

The use of schedules, apart from the constant work of mentor and mentoree, is vital to accomplish such a complex goal such as the presentation and defense of an FDP. Thus, meeting different tasks or objectives during this process will encourage the student to accomplish the whole process, assuring a good FDP performance as enough time is provided for the normal development of the project, according to our previous experience.

As it can be observed in this example, the development of an FDP can take up to 10 or 11 months, depending on the context of the student. As previously explained, RBL requires time so that the students get used to scientific tasks, research decisions, etc. [14]. Thus, if the mentoree is exclusively devoted to the FDP, without any pending subject in their degree, the schedule can be drastically reduced to around 5 or 6 months. Other factors affecting the duration of the FDP are mainly the difficulty in the laboratory work (including sample preparation, characterization and treatment) and the writing and review of the final degree report. Further, the FDP defense should be carried out in specific academic periods established by our university, which can delay the duration of the whole process.

In order to avoid delays, weekly meetings are usually held to check the progress made by the student.

*2.3. Training*

During the training stage, two main aspects are covered, such as general and specific issues. Among the former, the main aspects covered were the following:

- Safety at laboratories. This is the first and one of the most important subjects covered in this laboratory. Thus, a short guide or manual has been published in previous works, where many aspects are included, such as emergency and evacuation plans, suitable laboratory procedures or the use of personal protective equipment, among others [20]. In that sense, the first interview with the students could guide us to customize the information provided to them, as some information about the circumstances of the students (such as previous diseases) could alter the normal development of the work in the laboratory. Consequently, the implementation of this short safety course is important due to two reasons: the assurance of welfare as well as an education that can be useful for them in their professional future.
- Basic scientific education, including the proper use of scientific English, nomenclature or literature search (already mentioned in previous sections).
- Critical thinking enhancement, by means of the association of the current world situation to the topic of their research.

Regarding the specific training, the main laboratory skills are explained to students, including both general laboratory works (such as simple volumetric determinations, filtration processes, the use of micropipettes, etc.) and specific procedures related to their final degree project, such as the use of the oven for combustion or pyrolysis tests, gas chromatography for fatty acid methyl ester determination in biodiesel, etc. For this purpose, the student can resort to both the explanations of teachers or laboratory technicians as well as the use of laboratory guides for each task (including previous FDP or thesis, as well as international standards), working with samples that are not included in their final degree project (in order to avoid that valuable samples can be ruined during their learning process). As explained in Figure 2, during this stage, the supervision of the mentor decreased as the skills and self-confidence of the student improved, with the subsequent freedom of choice.

*2.4. Experimentation*

Once the student has the minimum laboratory skills, the experimentation stage takes place. As it happened during the whole process, the freedom of choice for FDP students was considerable because, as they gained experience, the control or supervision by their mentors was less and less noticeable. During experimentation, the following criteria are applied:

- There is a clear and unequivocal labeling system of the samples (treated or analyzed). Most of these samples were treated by one person, obtained from a raw material and treated in a certain way. Therefore, these details were reflected in labeled samples, as shown in this example: MS_R_1, where MS represents the initials of the student (Name and surname), R represents the sample (rapeseed, in this case) and the number is related to an experiment, which is related to a set of experimental conditions (clearly registered by the student).
- Apart from the suitable laboratory practices for each experiment, a specific order according to the schedule for each sample was carried out to meet all the specific deadlines for these tasks.
- All data and observations are registered by the student and the experiments were conducted in triplicate, which implies subsequent data treatment for the preparation of the final degree report.

### 2.5. Writing and Rehearsal

Once the student is in an advanced stage of the experimental process, the preparation of the final degree report and the subsequent speech or exhibition can be addressed. In that sense, different stages were covered, as follows:

- After the initial training concerning writing (covering scientific language, among other main aspects), a specific meeting was held to establish the main aspects concerning writing a scientific or technical report.
- Afterwards, the student started to write the final report, according to mentor's requirements and those established by the evaluators (see Table 3). During this process, once the student finished a specific section, it was reviewed by the mentor, adding comments or remarks to improve the quality of the final text. At this stage, the supervisor can help the student gain intuition skills.
- A similar process was followed with the final presentation and once it was reviewed by the mentor, the rehearsal process started.
- In order to rehearse the speech, initial online meetings were held with the mentor. This fact was especially important during the coronavirus outbreak, but it is also important, for instance, when the student lives far from our university (trying to promote active learning and engagement through online learning [67,68]).
- Finally, a series of general rehearsals are carried out "in situ", once the speech is depurated and well prepared by the student, to be adapted to the place of the final degree defense and feel more comfortable during the final performance.

**Table 3.** Main requirements and rating for FDPs.

| Requirement | Details |
|---|---|
| ECTS credits | 120 |
| Total time | 300 h |
| Maximum size | 100 pages [1] |
| Font style and size for titles | Arial Rounded MT Bold, 18 |
| Font style and size for main text | Arial, 11 |
| Content [1] | Introduction, materials and methods, results and discussion, economic study, conclusions and references |
| Presentation time | 30 min |
| **Evaluation criterion** | **% Final score** |
| Adequation to format | 10 |
| Content of the report | 50 |
| Oral presentation | 25 |
| Response to questions | 15 |

[1] For technical and research works.

### 2.6. FDP Defense

During the whole research process, we encourage the student not to forget the main objective of their work: the successful approval of the FDP and the subsequent university degree. For this reason, the main requirements and evaluation criteria are strictly taken into account in the first stages, mainly due to two reasons: first, to avoid extra work by changing formats or adapting the results; second, to adapt the student for a hypothetical publication process, which shows some similarities, showing them another aspect of the research profession. Taking these aspects into account, mentors will never allow a student to defend an FDP without complying with these minimum requirements, in order to avoid failures that would complicate the final degree approval by the evaluators.

It should be noted that these stages are not isolated in time, overlapping one another. For instance, the student can carry out some experiments while, at the same time, he/she can analyze or write some results for the final report. In fact, this overlapping usually

takes place when the student has more freedom, that is, at advanced stages, depending on their effort.

At the end of the FDP process (just before its defense), the students were asked to fill in a simple satisfaction survey to check their experience. For its completion, confidentiality and anonymity were assured as the survey was carried out without the presence of tutors (not including names or other compromising data) and all the surveys were opened during the last course included in this study.

The main questions included in this survey were the following:

- Did you enjoy your experience in the laboratory/mentoring process?
- Are you proud of the research work you have carried out?
- Would you consider a future scientific career?

For each question, the scores were the following: absolutely, 10; a lot, 7.5; more or less, 5; not much, 2.5; not at all, 0.

## 3. Results and Discussion

Once the defense of the final degree project was carried out and, therefore, all the research work was conducted, many conclusions were reached. The main outcomes of the FDP experience for our students covered their satisfaction (obtained from the survey), final grade (obtained during the FDP defense) and possible publication of their results (based on articles or conference communications).

Concerning the answers of the FDP students, most of them enjoyed the research process, that is, both the mentoring process and the laboratory work (8.8 out of 10), being quite proud of their research work (8.1 out of 10). However, they hardly seriously consider a future scientific career in general (4.8 out of 10), justifying that they are more interested in other professional careers.

Regarding the academic results and research publication, average grade obtained was high (8.4 out of 10), resulting in a good performance of the student throughout the whole research process. Moreover, as commented earlier, the students' satisfaction was equally considerable (8.8 out of 10). Regarding the scientific production, all the FDPs (100%) had enough material to publish, at least a conference communication, whereas 40% of the FPDs implied an article publication. Indeed, some of these new research works, such as [57], were automatically added to the recommended literature for new students (see Table 2), reflecting the importance of the contribution of students to the future development of other FDPs and the good functioning of the research work in our laboratory. Compared to previous students, who carried out their FDP between 2006–2007 and 2013–2014 courses (where this methodology was not implemented), there were clear differences when it comes to scientific production. Thus, only 17.6% of the FDPs had enough scientific quality to be included in scientific articles, whereas only 55% of these works were published as conference communication. In that sense, the implementation of this research-based course had a positive effect on scientific publication (doubling the possibilities of publication of these FDPs), as the quality of these works was high enough to be easily submitted to peer-reviewed systems, such as scientific journals or international conferences. As will be explained later, not only students benefited from this experience (which is the main purpose of this study) but also mentors (or teachers) and the department (and the institution).

After our professional relationship in the laboratory, we tried to maintain contact with postgraduate students, if possible. This way, their professional assignment could be assessed, as is shown in Figure 5. Although the sample of students is not very representative, some conclusions can be drawn. As observed, most of the students are currently working in scientific or technical jobs (46%), whereas 27% of students are devoted to other jobs that do not have anything to do with their career or are studying a master's degree, to apply for public jobs (or they are unemployed). Nevertheless, 18% of students are involved in research projects (in universities or research centers), although most of them on short-term contracts. This ratio is considerable, taking into account the situation in our region, where the number of university students currently participating in PhD programs

is below 10%, according to national data [8]. In this case and possibly due to the fact that students took an active part in their FDP, they gained enough research skills (such as critical thinking, problem-solving skills, data analysis or communication) to feel motivated to join PhD programs [69]. Equally, 9% of students are on training contracts (about different disciplines), which are equally unstable. Taking into account the scarcity of Engineers who study a PhD degree (although in an upward trend), the percentage found in this study was not low. Therefore, as other authors have pointed out, to avoid negative perceptions about the instability related to scientific careers (the so-called "route to the unknown"), it is vital to promote successful learning, teaching and research environments to produce working–learning friendly environments [70].

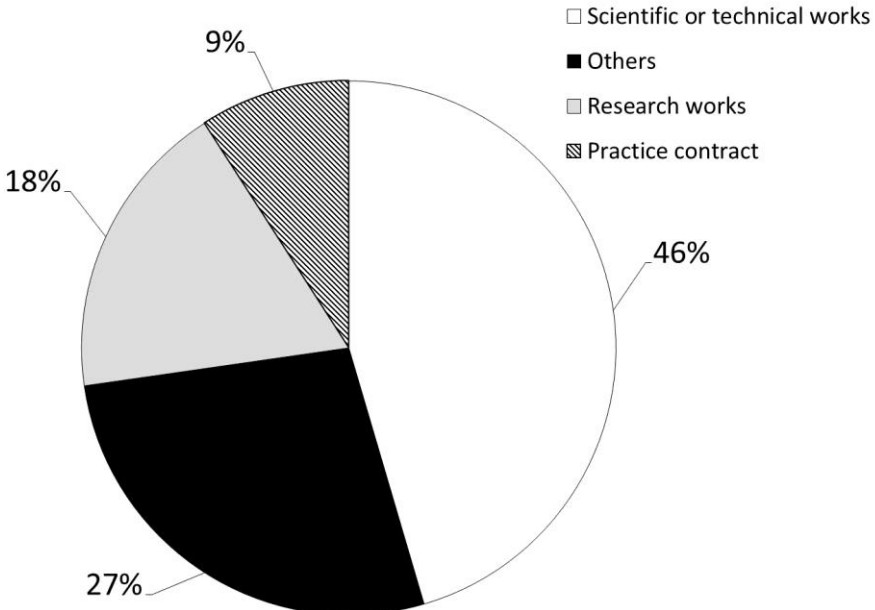

**Figure 5.** Jobs and occupations of graduate students.

As inferred from these data, there was a mutual benefit between FDP students and their teachers or mentors. From our experience, the main benefits of final degree projects are the following (Table 4):

**Table 4.** Main benefits for academic tutors and students during the FDP.

| Benefits for Academic Tutors | Benefits for Students |
| --- | --- |
| Publishing | Publishing |
| Database improvement | Professional skills |
| Laboratory assistance | New professional experience |

Thus, regarding mentors, the development of an FDP can be a good tool to improve many aspects of their research work, such as laboratory assistance to increase their experimental efficiency, improvement in databases to create an extensive collection of articles and references and facility to publish research. Concerning students, they get their corresponding degree thanks to this final educational stage, which also helped them to acquire basic professional skills related to laboratory works, offering another professional alternative. In that sense, their range of possibilities widened, helping them to make a better decision when looking for their suitable professional choice.

The procedure explained in this work was general and specific. In the first case, for STEM disciplines, these general instructions are useful (for instance, scientific English is always recurring in these disciplines). In the second case, the specific characteristics

of this experience can be easily adapted to every discipline at this educational level (by adapting the references corresponding to their research field, for example). Nevertheless, the foundations of this work and the logical sequence followed can be applied and adapted to other subjects, as other authors have recommended with their corresponding educational works related to problem-based or research-based learning [18].

To sum up and trying to answer the questions included in Section 1, this method contributed to the involvement of students, as their confidence and satisfaction were high, increasing the ratio of students devoted to research (in PhD programs) compared to regional data. Consequently, their engagement towards research was noticeable. Further, there were other benefits for students, such as new job alternatives, high academic results and the capability of carrying out high-quality research. On the other hand, the department improved the publishing ratio derived from these FDP, both in scientific journals and in international conferences, facilitating the routine in our laboratory. Finally, and according to the literature, this method shares the same foundations compared to other works, obtaining equivalent results when it comes to student satisfaction and engagement, but, in this case, the research-based program was different, adapted to a specific field that is not broadly covered in the literature, such as chemical and technical engineering.

## 4. Conclusions

The main findings of this experience were the following:

- A brief guide about "how to research" according to research-based learning was applied to a chemical engineering laboratory, which showed various advantages for both teachers and students, proving the mutual benefit of this educational experience.
- From teachers' point of view, this guide helped them to teach the main tasks related to final degree projects, as it is a tool that goes from general issues to specific ones, adapted to "real-life" situations in our laboratory. Nevertheless, this guide can be perfectly customized for other fields or subjects.
- Regarding FDP students, this guide facilitated their introduction to research and scientific methods in practice, which helped them widen their range of possibilities in their professional future. In addition, new skills and learning capabilities are offered, some of them general/transversal and other specific ones, which can help them in their future career.
- High satisfaction and engagement were found, for the tutor and the student, with the use of this didactic guide for research. However, and even though the ratio of students devoted to research was higher than at a regional level, students are not motivated enough to continue with this professional career, in general, probably because of the lack of public grants. Consequently, few students decide to continue with research work (with a PhD or a scientific contract).
- One of the challenges during FDPs is achieving results (conference communication or article publication) as soon as possible, in order to encourage the students to develop a scientific career. However, as the publication process is almost equivalent to the final degree projects, it is hardly ever possible to fulfill this point before the student finishes this level.
- Nevertheless, the effectiveness of the research work carried out in FDPs is high, with a high percentage of published articles and conference communications. This was due to the high quality of the research works, obtaining high rates in general.
- This work presents some challenges and further studies should be carried out, especially broadening the sample of studies (as the sample of teachers and students are not representative enough to extrapolate these results to universities as a whole or other educational contexts), which, in our case, is difficult due to the specific subject of the final degree projects and the long time that an FDP requires. Moreover, the specific design could not be suitable in other fields, which should be considered for further studies and this study did not focus on specific characteristics of students, such as gender or social context.

**Author Contributions:** Conceptualization, S.N.-D. and S.R.S.; methodology, S.N.-D. and S.R.S.; resources, J.M.E.M.; writing—original draft preparation, S.N.-D.; writing—review and editing, S.N.-D., S.R.S and J.M.E.M.; visualization, S.R.S. and J.M.E.M.; supervision, S.N.-D. and J.M.E.M.; project administration, J.M.E.M.; funding acquisition, J.M.E.M. All authors have read and agreed to the published version of the manuscript.

**Funding:** This research was funded by "Junta de Extremadura" and FEDER "FondosEuropeos de Desarrollo Regional" for the financial support received (grant numbers GR18150 and IB18028).

**Institutional Review Board Statement:** Not applicable.

**Data Availability Statement:** Not applicable.

**Acknowledgments:** The authors would like to thank the "Junta de Extremadura" and FEDER "Fondos Europeos de Desarrollo Regional" for the financial support received (grant numbers GR18150 and IB18028). We also would like to thank Isabel Vargas González, for her assistance and support, and our students, who always are teaching us important lessons at all levels.

**Conflicts of Interest:** The authors declare no conflict of interest.

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
