# Peer review of "Teaching How to Research: A Case Study on Chemical and Industrial Engineering Degrees"

_education, doi:10.3390/educsci12100673_

Round 1
Reviewer 1 Report
While this paper attempts to put forth results related to the student experience in completing a final degree project, it is difficult to determine what exactly happened in this study or what the implications are.
This paper is awkwardly organized and has extraneous and unnecessary figures and information (such as figure 2, table 2, figure 5, and figure 7). Some figures are unclear and unexplained (such as figure 3) while others are wholly unprofessional (figure 8).
It is unclear how this represents an educational research project and not just a report on an activity that takes place in this department. Without clarity of the research questions, design of the project, who the students and mentors are, or whether they were undertaking their own projects or contributing to an ongoing lab, it is difficult to recommend this paper for publication.
Author Response
Dear reviewer,
First of all, thank you for your review and advice, which has helped us to improve the quality of our work. We have tried to answer your questions and concerns, as follows:
While this paper attempts to put forth results related to the student experience in completing a final degree project, it is difficult to determine what exactly happened in this study or what the implications are.
We have clarified the results obtained, pointing out the main implications in a brief summary at the end of the “Results and Discussion” section, as follows (changes in red):
To sum up, and trying to answer the questions included in the introduction section, this method contributed to the involvement of students, as their confidence and satisfaction was high, increasing the ratio of students devoted to research (in PhD programs) compared to regional data. Consequently, their engagement towards research was noticeable. Also, there were other benefits for students, like new job alternatives, high academic results and the capability of carrying out high-quality research. On the other hand, the department improved the publishing ratio derived from these FDP, both in scientific journals and in international conferences, improving the routine in our laboratory. Finally, and according to the literature, this method shares the same foundations compared to other works, obtaining equivalent results when it comes to student satisfaction and engagement, but in this case the research-based program was different, adapted to a specific field that is not broadly covered in literature, like chemical and technical engineering.
Additionally, we have extended the discussion with further explanations and comparison with other works related to research-based learning, including further research data.
This paper is awkwardly organized and has extraneous and unnecessary figures and information (such as figure 2, table 2, figure 5, and figure 7). Some figures are unclear and unexplained (such as figure 3) while others are wholly unprofessional (figure 8).
We have improved the organization of the text, specifying each section and its purpose, and some figures and tables have been removed, explaining their content in the final text. We have properly explained Figure 3 (now Figure 2). The changes (in red) were as follows:
Based on the information gathered at this first meeting for our students (for eight consecutive courses, included in Table 1), their skills and personal circumstances are evaluated. They were asked about different aspects that we consider important to carry out a FDP properly, including the following items: References (for instance, how to cite an author), English level, writing and laboratory skills and others (where the students can add any particularity that can affect the correct development of their FDP). From our results, the main aspects that we should focus on (to carry out the final degree project properly) can be drawn out in Figure 2, where the percentage of students having problems with each subject is included.
Equally, Figure 8 has been removed, explaining its main findings.
It is unclear how this represents an educational research project and not just a report on an activity that takes place in this department. Without clarity of the research questions, design of the project, who the students and mentors are, or whether they were undertaking their own projects or contributing to an ongoing lab, it is difficult to recommend this paper for publication.
Thank you for your comments! We have specified these aspects in the final text (in . In this case, it is a research project related to biodiesel and biolubricant production, along with the energy use of biomass, contributing to the possible implementation of a biorefinery in our region. In this project, apart from the main technical and scientific results obtained (which can be found in the literature), there was a continuous interest in the involvement of students at university level, mainly through final degree programs. Thus, these students (whose characteristics are specified in the text) had their own project, taking part in this research project. The mentors were teachers, professors and technicians who took part in the main research project. More details about the project, staff and students were included in the final text, in Context and Methodology section, as follows (changes in red):
The staff includes professors, university teachers and scientific staff, who can act as mentors in final degree projects (FDP). Thus, there are two professors, three university teachers and five researchers (although professors and teachers also contribute to research), who usually participate in pairs for each FDP. All of them have a considerable experience in research (especially professors), with at least a two-year stay in this laboratory. All the FDPs taking place in this laboratory took part of a research project devoted to biofuels, including subjects such as biomass, biodiesel and biolubricant production. Thus, the main purpose of this project is to foster renewable energies in our region, but there is a specific part whose aim was to foster research at university level, making students a part of this project. That is the origin of this work, as we tried to assess the impact of research-based learning applied to renewable energies on future chemical and technical engineers. This way, the FDP of each student was part of a more complex project, but it was (in essence) their own work, gaining as much autonomy as they could.
- The main subjects explained in this branch of the department are related to Chemical and Technical Engineering, especially devoted to reaction mechanisms and kinetics during conversion reactions, in the context of energy and fuels.
- Apart from the abovementioned research project, the main research lines are related to energy from biomass, biofuels (mainly biodiesel) and bioproducts (such as glycerol or biolubricants), apart from catalyst characterization, among others.
In this context, one of the main tasks carried out in our laboratory is the development of final degree projects (FDP) for Chemical and Technical Engineering students. In this study, 16 male and 16 female students were included, whose age ranged from 22 to 28 years, depending on their easiness to complete their corresponding degrees. This group of students was compared with previous students who did not enjoy this methodology (from 2006-2007 to 2013-2014 academic year, 21 male and 14 female students, whose age range was 23-26 years).
Also, research questions were included in the introduction section.
Again, thank you for your comments, and we hope that our answers and changes can meet your requirements. Sincerely,
Reviewer 2 Report
The paper "Teaching how to research: a case study on Chemical and Industrial Engineering Degrees" is a description of a very successful Final Degree Project (FDP), a course at the end of a study program.
The FDP is described in detail. The authors write: "The aim of the work is to propose a methodology (mainly based on our own experience) to make research in engineering subjects more attractive for final degree students, in order to promote this discipline as a feasible career apart from the usual technical professional careers."
The authors recommend their methodology based on the high average rating of students, the result that all FDPs had produced at least enough material for a conference communication, and the high response score to two questions asked to students: how much they enjoyed the experience and how proud they are of their own work. The authors also followed their students after graduation.
The description of the FDP can be very interesting for lecturers from other STEM universities and it can be a great source of inspiration. It is nice to see that the authors have improved the quality of education in their institution in this way. However the pedagogical value of this case study is this stage not yet enough to be published in any pedagogical (research) journal.
The authors have defined the aim of the work, which is to "propose a methodology" but otherwise no research question has been set and/or answered about this proposed methodology. The research data that authors have presented is also very limited.
The case discussed is about research based learning. This is a very relevant topic in education, in particular higher STEM education. However, no context for this pedagogical approach nor the pedagogical research context on research based learning is provided in this article. There are 7 references on the list that support the argument that research in general is important for a country and that knowing how to do research is important for one's future career. More than 45 references are provided that refer to the content of the course - to the material intended for their students.
On the other hand, there are no references regarding the pedagogy of the course, nor are there any references to previous research in this area. In short, no context is given here. This means that the relevance and quality of the presented methodology is only argued based on own experience of the authors once and is not supported by any literature about the pedagogy of the research based learning or the research on it in STEM.
This way the contribution of this paper to the academic scholarship is insufficient and this needs to be improved to publish this work! A sound pedagogical context with relevant literature should be included which will support the proposed methodology used for FDP. It is necessary to explain what is new or what is specific in their approach and its success.
Author Response
Dear reviewer,
Firstly, thank you for your dedication, which has helped us to improve the quality of our work. We have tried to answer your questions and concerns, as follows:
The paper "Teaching how to research: a case study on Chemical and Industrial Engineering Degrees" is a description of a very successful Final Degree Project (FDP), a course at the end of a study program.
The FDP is described in detail. The authors write: "The aim of the work is to propose a methodology (mainly based on our own experience) to make research in engineering subjects more attractive for final degree students, in order to promote this discipline as a feasible career apart from the usual technical professional careers."
The authors recommend their methodology based on the high average rating of students, the result that all FDPs had produced at least enough material for a conference communication, and the high response score to two questions asked to students: how much they enjoyed the experience and how proud they are of their own work.
The authors also followed their students after graduation.
The description of the FDP can be very interesting for lecturers from other STEM universities and it can be a great source of inspiration. It is nice to see that the authors have improved the quality of education in their institution in this way. However the pedagogical value of this case study is this stage not yet enough to be published in any pedagogical (research) journal.
Thank you for your encouraging words, we will try and improve the pedagogical value of this case study by answering your questions and making the corresponding changes.
The authors have defined the aim of the work, which is to "propose a methodology" but otherwise no research question has been set and/or answered about this proposed methodology. The research data that authors have presented is also very limited.
We have specified the methodology, including the questions that has been set, as follows (changes in red):
Considering the above, the aim of this work was to propose a methodology (mainly based on our own experience, but also considering previous works) to make research in engineering subjects more attractive for final degree students, in order to promote this discipline as a feasible career apart from the usual technical professional careers. Also, preliminary results about the satisfaction and career opportunities of the students were included. Thus, we will try to answer the following questions:
- How can this method contribute to a higher commitment to research?
- Did this work improve the results dissemination of our department?
- What is the difference between this method and the previous ones concerning research-based learning?
Regarding the research data, we can not enlarge the number of students, as final degree programs are not abundant in our department. That is the reason why this project took place for years. Nevertheless, former students were included (who did not enjoy this methodology) to compare some results with our group of students, especially concerning scientific publication per FDP.
The case discussed is about research based learning. This is a very relevant topic in education, in particular higher STEM education. However, no context for this pedagogical approach nor the pedagogical research context on research based learning is provided in this article. There are 7 references on the list that support the argument that research in general is important for a country and that knowing how to do research is important for one's future career. More than 45 references are provided that refer to the content of the course - to the material intended for their students.
On the other hand, there are no references regarding the pedagogy of the course, nor are there any references to previous research in this area. In short, no context is given here. This means that the relevance and quality of the presented methodology is only argued based on own experience of the authors once and is not supported by any literature about the pedagogy of the research based learning or the research on it in STEM.
This way the contribution of this paper to the academic scholarship is insufficient and this needs to be improved to publish this work! A sound pedagogical context with relevant literature should be included which will support the proposed methodology used for FDP. It is necessary to explain what is new or what is specific in their approach and its success.
We have added more references, improving the quality of the reasoning and the global quality of the article, focusing on research based learning on STEM education. Thus, the pedagogical context was improved, supporting our proposed methodology and explaining what is new or specific in our approach. All these details are included in the introduction section (regarding research-based learning and our approach) and Context section (where the pedagogical context was improved by adding more details. Also, the results were compared and supported with further works.
Again, thank you for your remarks and kind words, and we hope that our answers and changes can meet your expectations. Sincerely,
Round 2
Reviewer 2 Report
Thank you for improving your paper, including the scientific context and research questions. This is an interesting and relevant study for the people who work in higher education STEM disciplines.